# Characterization and Relaxation Properties of a Series of Monodispersed Magnetic Nanoparticles

**DOI:** 10.3390/s19153396

**Published:** 2019-08-02

**Authors:** Yapeng Zhang, Jingjing Cheng, Wenzhong Liu

**Affiliations:** 1School of Artificial Intelligence and Automation, Huazhong University of Science and Technology, Wuhan 430074, China; 2Key Laboratory of Image Processing and Intelligent Control (Huazhong University of Science and Technology), Ministry of Education, Wuhan 430074, China

**Keywords:** magnetic nanoparticles, contrast agent, relaxation, relaxation rate, Langevin model, magnetic field inhomogeneity

## Abstract

Magnetic iron oxide nanoparticles are relatively advanced nanomaterials, and are widely used in biology, physics and medicine, especially as contrast agents for magnetic resonance imaging. Characterization of the properties of magnetic nanoparticles plays an important role in the application of magnetic particles. As a contrast agent, the relaxation rate directly affects image enhancement. We characterized a series of monodispersed magnetic nanoparticles using different methods and measured their relaxation rates using a 0.47 T low-field Nuclear Magnetic Resonance instrument. Generally speaking, the properties of magnetic nanoparticles are closely related to their particle sizes; however, neither longitudinal relaxation rate $r_{1}$ nor transverse relaxation rate $r_{2}$ changes monotonously with the particle size d. Therefore, size can affect the magnetism of magnetic nanoparticles, but it is not the only factor. Then, we defined the relaxation rates $r_{i}^{\prime }$ (*i* = 1 or 2) using the induced magnetization of magnetic nanoparticles, and found that the correlation relationship between $r_{1}^{\prime }$ relaxation rate and $r_{1}$ relaxation rate is slightly worse, with a correlation coefficient of $R^{2}$ = 0.8939, while the correlation relationship between $r_{2}^{\prime }$ relaxation rate and $r_{2}$ relaxation rate is very obvious, with a correlation coefficient of $R^{2}$ = 0.9983. The main reason is that $r_{2}$ relaxation rate is related to the magnetic field inhomogeneity, produced by magnetic nanoparticles; however $r_{1}$ relaxation rate is mainly a result of the direct interaction of hydrogen nucleus in water molecules and the metal ions in magnetic nanoparticles to shorten the $T_{1}$ relaxation time, so it is not directly related to magnetic field inhomogeneity.

## 1. Introduction

Magnetic iron oxide nanoparticles (MIONPs) have developed rapidly in recent years and have been widely used in biology, physics and medicine. They are quite small, usually nanoscale, and because of their scale, they can manifest many unique properties, such as superparamagnetism, i.e., when the applied magnetic field approaches zero, the induced magnetization and coercivity are also zero [1,2,3,4]. In addition, MIONPs have good temperature performance and can be used as temperature sensors. Some scholars used them to make some progress in the field of magnetic temperature measurement [5,6,7,8,9,10,11]. Due to their low toxicity, biocompatibility, and specificity after surface modification, MIONPs can be used as a target for drug delivery and disease treatment [12,13,14,15]. MIONPs are often used as contrast agents [16,17,18,19,20] in magnetic resonance imaging (MRI), which is one of the most important imaging methods in the field of medical diagnosis and scientific research [21,22,23,24]. MRI is a kind of imaging technology, which mainly uses the resonance effect of an electromagnetic field and hydrogen nucleus spin. Then, according to the collected magnetic resonance signal, imaging information from the tested object can be established. MRI not only has many imaging parameters and can be imaged in any direction without ionizing radiation damage to the human body, but can also perform non-invasive imaging of the internal structure or tissue of the human body or organism. According to the energy exchange between nuclear spin and the outside, hydrogen protons have two main relaxation mechanisms: longitudinal relaxation (spin-lattice relaxation) and transverse relaxation (spin-spin relaxation), which correspond to two relaxation time parameters, respectively: longitudinal relaxation time ($T_{1}$ relaxation time) and transverse relaxation time ($T_{2}$ relaxation time). These two relaxation mechanisms have a direct impact on the quality of magnetic resonance imaging. When the contrast agent (such as MIONPs) is added to the tested object, it will produce induced magnetization under the excitation of the magnetic field, which will affect the distribution and uniformity of the magnetic field around it, and then change the relaxation mechanisms of the protons around it; in short, the relaxation times will change. More intuitively, it will enhance the image contrast and speed up the image efficiency [23,25,26]. The enhancement effect of contrast agents on MRI can be expressed by the relaxation rate ($r_{1}$ relaxation rate and $r_{2}$ relaxation rate). Current research focuses on how to reduce proton $T_{1}$ relaxation time and $T_{2}$ relaxation time, how to improve the contrast performance of lesions and surrounding tissues, and how to accelerate the relaxation rate. In these studies, MIONPs can play a significant role. For example, different synthetic methods, size control, surface modification, etc. are used to improve their performance as a contrast agent. In addition, some scholars studied the use of different proportions of other metal materials such as Co, Zn and Mn doped iron oxide to enhance its saturation magnetization, thereby improving the relaxation enhancement properties of magnetic nanoparticles [27,28,29,30].

Characterization of the properties of magnetic nanoparticles plays an important role in the application of magnetic particles. In this paper, a series of commercial single core magnetic nanoparticles (SHP series, Ocean Nanotech, San Diego, CA, USA) with different nominal particle sizes (5 nm, 10 nm, 15 nm, 20 nm, 25 nm, 30 nm) were characterized using different methods. They can be stably monodispersed in aqueous solutions. Generally speaking, the properties of magnetic nanoparticles are closely related to their sizes, so we firstly measured the particle sizes using transmission electron microscopy and dynamic light scattering. Then the relaxation time of samples with different Fe ion concentrations was measured in a 0.47 T low-field nuclear magnetic resonance (LF-NMR) instrument and fitted to obtain $r_{1}$ relaxation rate and $r_{2}$ relaxation rate. However, neither $r_{1}$ relaxation rate calculated nor $r_{2}$ relaxation rate calculated changes monotonously with the particle size d. Because the addition of magnetic nanoparticles mainly changes the uniformity of the ambient magnetic field of the surrounding water molecules, and then changes the relaxation time, we further seek the relationship between the relaxation time and the induced magnetization of the magnetic nanoparticles in the magnetic field, and define the $r_{1}^{\prime }$ relaxation rate and $r_{2}^{\prime }$ relaxation rate. Because the $T_{2}$ relaxation process is very sensitive to the inhomogeneity of the magnetic field, and the addition of magnetic nanoparticles can directly affect the inhomogeneity of the magnetic field, it is found that the correlation between $r_{2}^{\prime }$ relaxation rate and $r_{2}$ relaxation rate is good. The effect of magnetic nanoparticle contrast agent on $T_{1}$ relaxation mechanism is mainly due to the direct interaction of the hydrogen nucleus in water molecules and metal ions in magnetic nanoparticles; therefore $T_{1}$ relaxation mechanism is not directly related to the magnetic field inhomogeneity. Therefore, the magnetic properties of magnetic nanoparticles are influenced by many factors, among which particle size is only one.

## 2. Materials and Methods

### 2.1. Magnetic Nanoparticles

We use an SHP series commercial magnetic nanoparticle reagent (Ocean NanoTech, San Diego, CA, USA), which is a single core magnetic nanoparticle. It is composed of ferric oxide magnetic nanoparticles, which are coated with carboxylic acid groups on the surface with good water solubility and dispersity. Its original iron ion concentration is 5 mg/mL. Six kinds of magnetic nanoparticle reagents were selected: SHP-05, SHP-10, SHP-15, SHP-20, SHP-25 and SHP-30 (nominal particle sizes are 5, 10, 15, 20, 25 and 30 nm, respectively, with a tolerance of < 2.5 nm). In addition, according to the measurement method of relaxation rate, samples with different Fe concentrations need to be prepared for each particle size of the magnetic nanoparticle reagent. Therefore, the magnetic nanoparticle reagent with the above particle size were diluted by deionized water to prepare five different Fe ion concentrations (1.79 mM, 1.12 mM, 0.89 mM, 0.45 mM, 0.22 mM), totaling 30 samples.

### 2.2. Transmission Electron Microscopy

The magnetic nanoparticles were imaged by transmission electron microscopy (TEM) (H-7000FA, HITACHI, Tokyo, Japan) with 110 kV to characterize their core size.

### 2.3. Dynamic Light Scattering

Dynamic Light Scattering (DLS) can detect the diffusion motion of particles and determine the hydrodynamic size (the overall size including magnetic core, polymer coating layer and the surrounding water layer) distribution of particles. Zetasizer Nano ZS90 (Malvern−Panalytical, Malvern, England) was used to measure the hydrodynamic size distribution of the above magnetic nanoparticles at a fixed scattering angle of 90 degree. The autocorrelation function of scattered light is analyzed, and the size distribution is calculated by assuming that the particle is an equivalent sphere.

### 2.4. LF-NMR

The relaxation time was measured using an LF-NMR instrument with a magnetic field of 0.47 T (MiniPQ001-20-15 mm, Niumag, Suzhou, China). The temperature of the main magnet and the sample chamber of the instrument is set at 35 °C To shorten the temperature balance time after the samples are put into the sample chamber, the samples are kept in an incubator set at 35 °C before the experiment. After starting the experiment, the samples were placed in the sample chamber of the LF-NMR instrument in turn for about 3 min, so that the temperature of the samples in the sample chamber could reach the thermal equilibrium as far as possible. The $T_{1}$ relaxation time is measured four times by using Inverse Recovery (IR) Sequence and then $T_{2}$ relaxation time was measured by using the CPMG (Carr-Purcell-Meiboom-Gill) sequence five consecutive times.

## 3. Results

### 3.1. Characterization of Magnetic Nanoparticles Samples: TEM, DLS

**TEM:** The TEM images of SHP-05, SHP-10, SHP-15, SHP-20, SHP-25 and SHP-30 magnetic nanoparticles samples in Figure 1 show that they are monodispersed magnetic nanoparticles.

**DLS**: The hydrodynamic size distributions are shown in Figure 2. To reduce the measurement error, each magnetic nanoparticle sample was measured three times in succession, and the statistical mean values were used.

Generally, we believe that the hydrodynamic size distribution of magnetic nanoparticles follows lognormal distribution [31,32,33,34]
(1)$$f\left(d\right)=\frac{1}{d\sqrt{2\pi }\sigma }exp\left[-\frac{1}{2\sigma^{2}}{\left(ln\frac{d}{\mu }\right)}^{2}\right]$$
here, d is the diameter of particles, μ is the median diameter of the lognormal distribution, σ is the standard deviation of lnd. Therefore, the lognormal distribution function is used to fit the hydrodynamic size distribution measured by DLS, and the results are shown in Table 1.

Dynamic light scattering (DLS) is a physical characterization method, which can be used to measure the particle size distribution of solutions or suspensions, and also to measure the behavior of complex fluids such as concentrated polymer solutions. The irregular random diffusion of magnetic particles in aqueous solution will attract some water molecules to move together on its surface, that is to say, water film is formed on its surface. It can be seen that the measurement results seem to be somewhat abnormal and have little correlation with the nominal core sizes. The results of DLS measurements using the same magnetic particles as us are not entirely consistent in other studies [35,36,37]. This may be caused by measurement errors, methods, etc., or more complex underlying causes.

### 3.2. Waiting Time Dependence of $T_{2}$ Relaxation Time

The relaxation time of each magnetic nanoparticle samples was then measured using a 0.47 T LF-NMR instrument. In addition, it was reported in much of the literature that the relaxation time (especially $T_{2}$ relaxation time) of magnetic nanoparticle aqueous solution samples obtained from NMR measurements is time-dependent [38,39,40,41]. That is to say, the measured relaxation times are related to the waiting time that the sample undergoes after being put into the sample chamber of the NMR instrument. In our experiment, because each sample was kept for about 3 min in the sample chamber, then the $T_{1}$ relaxation time was measured several times, and then the $T_{2}$ relaxation time was measured several times in succession. Therefore, depending on the above-mentioned articles on the waiting time dependence of $T_{2}$ relaxation time, this paper did not set the waiting time accurately. However, in order to illustrate the problem as much as possible, we define the waiting time $t_{w}^{\prime }$, and set the waiting time $t_{w}^{\prime }$ for the first measurement of the $T_{2}$ relaxation time of each sample to 0 s. The waiting time $t_{w}^{\prime }$ of the same sample for subsequent $T_{2}$ relaxation time measurement was set according to the time interval from the first measurement. Generally, we choose the $T_{2}$ relaxation time measurement data of the sample with the highest Fe concentration for each particle size, and plot the relationship between $T_{2}$ relaxation time and waiting time $t_{w}^{\prime }$, as shown in Figure 3 below.

It can be clearly seen that the $T_{2}$ relaxation time of the magnetic nanoparticles used in our experiment did not show dependence on $t_{w}^{\prime }$ waiting time. The main reason may be that the sizes of magnetic nanoparticle samples are relatively small, and surface-coated carboxylic acid groups produce electrostatic repulsion, which makes it possible for them to disperse stably in water phase without agglomeration under the effect of external magnetic field. Of course, it is also possible that the magnetic field (0.47 T) of the LF-NMR instrument in this paper is too low to make the magnetic nanoparticles in the magnetic field cluster into chains.

Regardless of the reason, the extrapolation method in these references is not needed in this paper [38,39,40,41], but the $T_{1}$ relaxation time and $T_{2}$ of relaxation time measured can be directly used to obtain the relaxation rate information of corresponding samples.

### 3.3. Relaxation Rate

In MRI, longitudinal relaxation rate $r_{1}$ and transverse relaxation rate $r_{2}$ are the main indicators to evaluate the enhancement effect of magnetic nanoparticles as contrast agent. Relaxation rate, in units of mM^−1^s^−1^, is the inverse of $T_{1}$ or $T_{2}$ relaxation time of protons when the concentration of Fe ion in magnetic nanoparticle dispersion solutions is 1 mM [39,42,43]:(2)$$1/T_{1,sup}=1/T_{1,water}+r_{1}c_{\mathrm{Fe}}$$
(3)$$1/T_{2,sup}=1/T_{2,water}+r_{2}c_{\mathrm{Fe}}$$
where $T_{1,sup}$ and $T_{2,sup}$ are the $T_{1}$ and $T_{2}$ relaxation time of magnetic nanoparticle suspension solution respectively; $T_{1,water}$ and $T_{2,water}$ are the intrinsic $T_{1}$ and $T_{2}$ relaxation time of water respectively; $c_{\mathrm{Fe}}$, in the unit of mM (mmol/L), is the concentration of Fe ion in magnetic nanoparticle suspension solution.

According to the above definition, we can simply prepare several magnetic nanoparticle water solution samples with different Fe ion concentrations, and use NMR instrument to measure their relaxation time respectively. Then taking $T_{i,water}$ (*i* = 1 or 2) and $r_{i}$ (*i* = 1 or 2) as fitting parameters, by fitting the curve between the inverse of relaxation time and the concentration of Fe ion, the slope is the corresponding relaxation rate, as shown in Figure 4.

To make it more intuitive, we summarized the relaxation rate information of SHP series magnetic nanoparticle samples, as shown in Figure 5.

The addition of contrast agent will affect $T_{1}$ and $T_{2}$ relaxation time, but the degree of influence on the two relaxation times may be different, that is to say, $r_{1}$ relaxation rate and $r_{2}$ relaxation rate may be different, so we can simply classify the contrast agent using $r_{2}$ to $r_{1}$ ratio, i.e., $r_{2}/r_{1}$ [25,44,45,46,47,48]. It is generally believed that when the ratio $r_{2}/r_{1}<2$, the contrast agent works better as $T_{1}$ contrast agent; when $2<r_{2}/r_{1}<10$, the contrast agent can work as both $T_{1}$ contrast agent and $T_{2}$ contrast agent, that is to say, it is $T_{1}$-$T_{2}$ dual mode contrast agent; when the ratio $r_{2}/r_{1}>10$, the contrast agent works better as $T_{2}$ contrast agent.

After calculating the $r_{2}/r_{1}$ ratio of the SHP series magnetic nanoparticles, and according to the above classification method, we then attempted to classify them, as shown in Figure 6.

It can be observed that the $r_{2}/r_{1}$ ratios of SHP-05, SHP-10, SHP-15, SHP-20 and SHP-25 magnetic nanoparticle samples are between 2 and 10, so they can be considered to be $T_{1}$-$T_{2}$ dual mode contrast agents. While the $r_{2}/r_{1}$ ratio of the SHP-30 magnetic nanoparticle sample is close to 20, it will work more suitably as $T_{2}$ contrast agent.

### 3.4. Analysis of Relaxation Rate

In addition, it can be found that neither $r_{1}$ relaxation rate calculated nor $r_{2}$ relaxation rate calculated changes monotonously with the particle size d [49,50]. When magnetic nanoparticles are added, the induced magnetization thereof will be produced under the exciting of the static magnetic field, which will change the ambient magnetic field, increase their inhomogeneity and shorten the relaxation time of the surrounding water hydrogen proton. This is also the basic principle of magnetic nanoparticles as contrast agents in MRI. It can be assumed that the magnetic field inhomogeneity is directly related to the induced magnetization produced by magnetic nanoparticles in magnetic field, and the induced magnetization can be described by the Langevin model [8,51,52,53,54]:(4)$$M=Nm\left(coth\left(\frac{mB}{kT}\right)-\frac{kT}{mB}\right)$$
where N is the number of magnetic nanoparticles per unit volume (i.e., the concentration of magnetic nanoparticles); m is the magnetic moment of a single magnetic nanoparticle; B is the static magnetic field; k is the Boltzmann constant and T is the absolute temperature.

It can be seen that the induced magnetization of magnetic nanoparticles system in a static magnetic field is related to the external excitation magnetic field, temperature, concentration of magnetic particles and magnetic moment of magnetic particles. Generally, magnetic nanoparticles are simplified to sphere shapes, so the magnetic moment m of magnetic nanoparticle can be expressed as the following equation:(5)$$m=M_{sat}V=M_{sat}\frac{\pi d^{3}}{6}$$
where, $M_{sat}$ is the saturation magnetization of magnetic nanoparticle. d is the diameter of magnetic nanoparticles.

In the previous calculation of the relaxation rate, the concentration of Fe ion is taken as a reference. Here we assume that the magnetic field inhomogeneity caused by the addition of magnetic particles is proportional to the induced magnetization of magnetic nanoparticles, so we define the relaxation rate by referring to the induced magnetization of each sample. that is to say, we define $r_{i}^{\prime }$ (*i* = 1 or 2) relaxation rate. The method for calculating the $r_{i}^{\prime }$ (*i* = 1 or 2) relaxation rate is similar to that shown in Equations (2) and (3), except that the induced magnetization M (unit A.m) at different magnetic particle concentrations can be used to replace the Fe concentration $c_{\mathrm{Fe}}$. When the induced magnetization intensity of MIONPs in water solution is 1 A.m, the inverse of $T_{1}$ relaxation time and $T_{2}$ relaxation time of the hydrogen proton is $r_{1}^{\prime }$ relaxation rate and $r_{2}^{\prime }$ relaxation rate respectively. It can be concluded that the unit of $r_{1}^{\prime }$ relaxation rate and $r_{2}^{\prime }$ relaxation rate is A^−1^m^−1^s^−1^.

Therefore, according to the Langevin model, combined with the actual relaxation time measurement experiment, the induced magnetization of each magnetic nanoparticle sample in LF-NMR instrument is simulated. The static magnetic field B = 0.47 T (the main magnetic field of LF-NMR instrument MiniPQ001-20-15 mm), temperature T = 308.17 K (the temperature of the sample chamber of the LF-NMR instrument MiniPQ001-20-15 mm). In much of the literature, the saturation magnetization $M_{sat}$ of SHP series magnetic nanoparticles was measured and involved, but the values measured (or assumed) are different, but the difference is not significant [35,55,56,57,58]. From Equation (5), it can be seen that magnetic moment m is proportional to the cube power of the particle size d, but is only proportional to the first power of saturation magnetization $M_{sat}$. In other words, the influence of magnetic particle size d on magnetic moment m is much bigger than the influence of the saturation magnetization $M_{sat}$ on magnetic moment m. Therefore, we can say that the magnetic moment m is mainly dominated by particle size d. Therefore, according to the above references, it is reasonable to assume that the saturation magnetic moments of SHP series magnetic nanoparticles are all the same, i.e., $M_{sat}$ = 4.5 × 10^5^ A/m.

In addition, the induced magnetization of magnetic particle system is also affected by the concentration of magnetic nanoparticles. Because the Langevin model needs concentration information from the magnetic nanoparticles, however the samples previously prepared are based on the concentration of Fe ion, so we need a conversion process. According to the Ocean Nanotech official website [59], the ratios of magnetic nanoparticle concentration to Fe ion concentration of SHP series magnetic nanoparticles, with different particle sizes, are different. After these conversion processes, finally, the induced magnetization M of different magnetic particle samples can be calculated using the Langevin model as shown in Equation (4). Then, by fitting the curve between the inverse of relaxation time and the induced magnetization M, the slope is the corresponding $r_{1}^{\prime }$ relaxation rate and $r_{2}^{\prime }$ relaxation rate, as shown in Figure 7. Moreover, the ratios of magnetic nanoparticle concentration to Fe ion concentration of every SHP reagent can be seen in Table A1 in Appendix A. In addition, parameters include magnetic nanoparticle concentration, induced magnetization, mean values and standard deviations of the measured relaxation time of the all tested magnetic nanoparticle samples are listed in Table A2 in Appendix A.

Similarly, in order to be more intuitive, we summarized the $r_{1}^{\prime }$ relaxation rate and $r_{2}^{\prime }$ relaxation rate information of SHP series magnetic nanoparticle samples, as shown in Figure 8.

It can be seen that the $r_{1}^{\prime }$ relaxation rate or $r_{2}^{\prime }$ relaxation rate still have little relationship with the particle size. Next, we analyze the correlation between the $r_{i}^{\prime }$ (*i* = 1 or 2) relaxation rate and the $r_{i}$ (*i* = 1 or 2) relaxation rate, and then linearly fit them, respectively, as shown in Figure 9. From the fitting results, it can be concluded that the correlation relationship between $r_{1}^{\prime }$ relaxation rate and $r_{1}$ relaxation rate is slightly worse, the correlation coefficient $R^{2}$ = 0.8939, while the correlation relationship between $r_{2}^{\prime }$ relaxation rate and $r_{2}$ relaxation rate is very obvious, the correlation coefficient $R^{2}$ = 0.9983.

The CPMG pulse sequence, designed for MRI and NMR, is designed to eliminate the influence of magnetic field inhomogeneity on the measurement of $T_{2}$ relaxation time as much as possible. Thus, we can obtain intrinsic $T_{2}$ relaxation time induced by spin-spin interaction. However, when magnetic nanoparticles are added to the aqueous solution sample, there are interactions between proton dipoles, proton dipoles and lattices, and between proton dipoles and electron dipoles of magnetic particles. These interactions will affect the relaxation mechanism of proton subsystems. In particular, magnetic nanoparticles will produce induced magnetization with the exciting of the magnetic field, which will change the magnetic field around them, aggravate the magnetic field inhomogeneity, accelerate the dephasing of hydrogen proton spin, and then change the relaxation of protons in the magnetic field. Moreover, the intensity of this impact will vary with the distance between hydrogen protons and magnetic particles. MIONPs with superparamagnetism can produce strong induced magnetization under the exciting magnetic field, which has a great influence on the inhomogeneity of the environmental magnetic field. Therefore, the $T_{2}$ relaxation time measured at this time includes all the above effects, and it can be said that it is mainly affected by the magnetic field inhomogeneity (the magnetic field inhomogeneity caused by the addition of magnetic nanoparticles) [60,61].
(6)$$\frac{1}{T_{2,observe}}=\frac{1}{T_{2}}+\frac{1}{T_{2}^{\prime }}=\frac{1}{T_{2}}+\gamma \Delta B$$
where $T_{2,observe}$ is the transverse relaxation time measured; $T_{2}$ is the intrinsic lateral relaxation time; $T_{2}^{\prime }$ is the transverse relaxation time due to the inhomogeneity of the magnetic field and can be expressed by γΔB, where γ is the gyromagnetic ratio of the hydrogen proton, ΔB is the inhomogeneity of the magnetic field in the sample system and can be given by the Langevin model. From Equation (6) we can see that the relationship between $T_{2,observe}$ relaxation time and induced magnetization of magnetic nanoparticles system is obvious, so the correlation between $r_{2}$ relaxation rate and $r_{2}^{\prime }$ relaxation rate is good.

## 4. Discussion

As can be seen in Figure 5, basically, the smaller the particle size of magnetic nanoparticles, the larger the $r_{1}$ relaxation rate. The effect of magnetic nanoparticle contrast agent on $T_{1}$ relaxation mechanism is mainly due to the direct interaction of hydrogen nucleus in water molecules and the metal ions in magnetic nanoparticles to shorten the $T_{1}$ relaxation time, which can be explained using the Solomon-Bloembergen-Morgan theory (SBM) [62,63], so it is not directly related to magnetic field inhomogeneity. In short, it is related to the number of coordination water molecules, the exchange rate of water molecules, the rotation time of complexes and so on. As far as the metal ions (Fe) in magnetic nanoparticles are concerned, they have unpaired electrons, which interact with water electrons, such as by coordinating. The smaller the particle size, the larger the specific surface area S/V, and the easier the interaction between unpaired electrons and water electrons, so the larger the $r_{1}$ relaxation rate is. As far as $r_{2}$ relaxation rate is concerned, it is directly related to the magnetic field inhomogeneity caused by magnetic nanoparticles in this paper. The magnetization induced by magnetic nanoparticles in the magnetic field can be considered to be directly affecting the magnetic field inhomogeneity. The induced magnetization of magnetic nanoparticles is directly related to the magnetic moment ($m=M_{sat}\frac{\pi d^{3}}{6}$). Therefore, if the saturation magnetization $M_{sat}$ of magnetic particles is constant, the larger the size of magnetic particles, the greater the induction magnetization and the larger the $r_{2}$ relaxation rate. Therefore, according to the respective influencing factors of $r_{1}$ relaxation rate and $r_{2}$ relaxation rate, it can be concluded that the larger the particle size, the larger the $r_{2}/r_{1}$, as shown in Figure 6. However, as far as the actual $r_{2}$ relaxation rate measured is concerned, it is basically shown that the larger the particle size is, the larger the $r_{2}$ relaxation rate is, but the data of SHP-20 and SHP-25 seem to be somewhat abnormal, as shown in Figure 5. The possible reason for this is that the saturation moments of SHP magnetic nanoparticles are not the same, and TEM imaging shows that the shape of SHP magnetic nanoparticles is not perfect spherical. Therefore, the magnetic properties of magnetic nanoparticles are affected by many factors, including composition, preparation process, surface coating, particle size, saturation magnetization and so on, while relaxation rate is only a simple characterization parameter. Moreover, Dynamic light scattering (DLS), which is a physical characterization method, and can be used to measure the particle size distribution of solutions or suspensions, and also to measure the behavior of complex fluids such as concentrated polymer solutions. The irregular random diffusion of magnetic particles in an aqueous solution will attract some water molecules to move together on its surface, that is to say, a water film is formed on its surface. The results of DLS measurements of magnetic nanoparticles used by us in different studies are not entirely consistent, which may be caused by their own measurement errors, methods and so on.

## 5. Conclusions

Magnetic nanoparticles are widely used as contrast agents for MRI. In this paper, SHP series magnetic nanoparticles (Ocean Nanotech) with different nominal sizes were selected for characterization experiments, and their relaxation rates were measured using a 0.47 T LF-NMR instrument. It was found that neither $r_{1}$ relaxation rate nor $r_{2}$ relaxation rate calculated changes monotonously with the particle size d. Size can affect the magnetic properties of magnetic nanoparticles, but it is not the only factor. There are other factors, such as morphology, agglomeration and so on. Furthermore, the induced magnetization of magnetic particle system in a static magnetic field can be calculated using the Langevin model, which serves as the source of magnetic field inhomogeneity in the measurement of relaxation time. Then the relationship between relaxation time and induced magnetization of magnetic nanoparticle samples in magnetic field is defined as $r_{1}^{\prime }$ relaxation rate and $r_{2}^{\prime }$ relaxation rate. The $T_{2}$ relaxation process is very sensitive to the magnetic field inhomogeneity, and the addition of magnetic nanoparticles can directly affect the magnetic field inhomogeneity. Therefore, it is found that the correlation between the $r_{2}^{\prime }$ relaxation rate and the $r_{2}$ relaxation rate is very good, and the correlation coefficient reaches 0.9983. $T_{1}$ relaxation mechanism is not directly related to the magnetic field inhomogeneity. The effect of the magnetic nanoparticle contrast agent on $T_{1}$ relaxation mechanism is mainly due to the direct interaction of the hydrogen nucleus in water molecules and the metal ions in magnetic nanoparticles, so the correlation coefficient between the $r_{2}^{\prime }$ relaxation rate and the $r_{2}$ relaxation rate is only 0.8939.

## Figures and Tables

**Figure 1 sensors-19-03396-f001:**
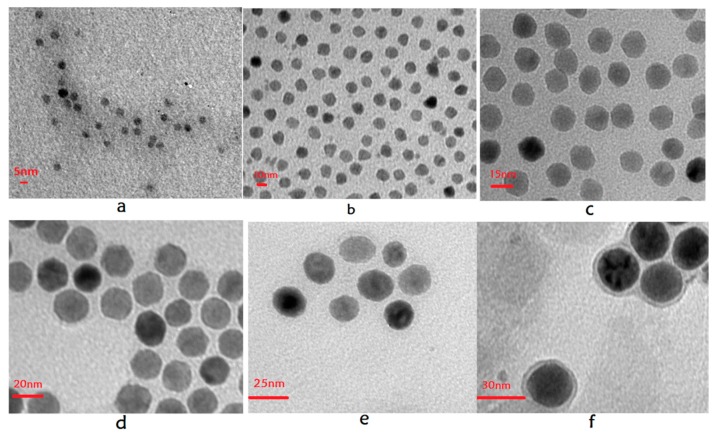
TEM images of SHP series magnetic nanoparticles samples. (**a**) SHP-05; (**b**) SHP-10; (**c**) SHP-15; (**d**) SHP-20; (**e**) SHP-25; (**f**) SHP-30.

**Figure 2 sensors-19-03396-f002:**
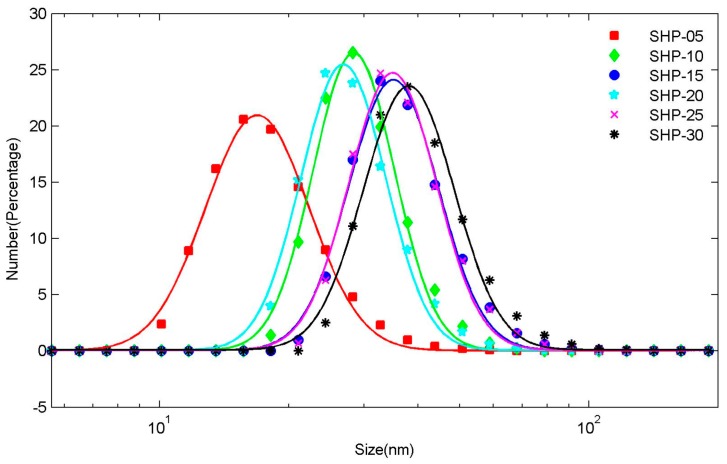
Hydrodynamic size distribution of SHP series magnetic nanoparticles. The discrete points are the measured hydrodynamic size distributions, and the solid lines are the fitting curve obtained using lognormal distribution.

**Figure 3 sensors-19-03396-f003:**
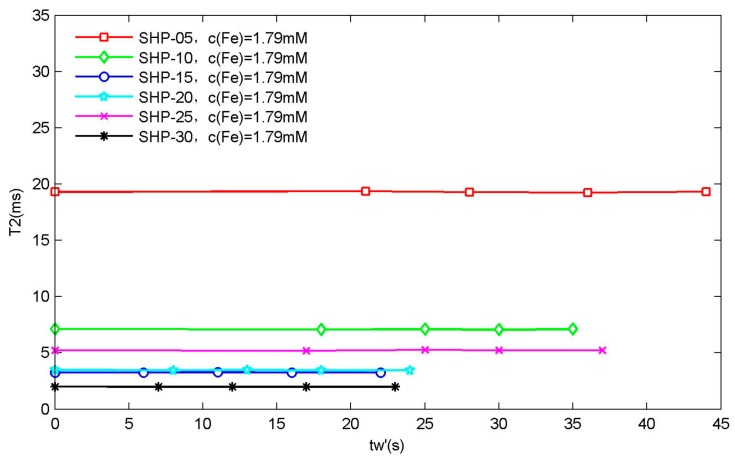
Waiting time dependence of $T_{2}$ relaxation time. It can be seen that the $T_{2}$ relaxation time of magnetic nanoparticle samples with different particle sizes hardly varies with the waiting time $t_{w}^{\prime }$ under the current test conditions.

**Figure 4 sensors-19-03396-f004:**
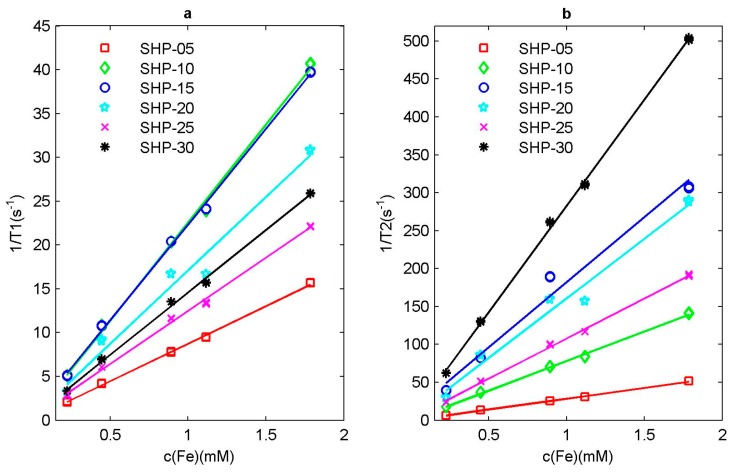
Relaxation rate of SHP series magnetic nanoparticle sample. (**a**) Inverse of longitudinal relaxation time $1/T_{1}$ and (**b**) inverse of transverse relaxation time $1/T_{2}$ with respect to Fe ion concentration $c_{\mathrm{Fe}}$.

**Figure 5 sensors-19-03396-f005:**
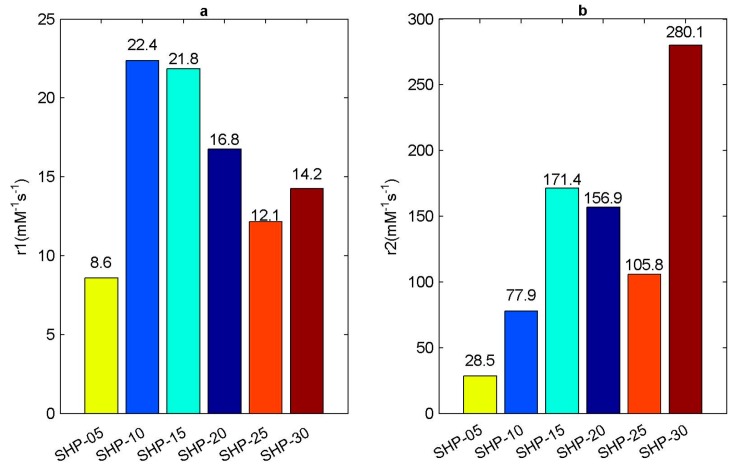
Relaxation rate of SHP series magnetic nanoparticle samples. (**a**) $r_{1}$ relaxation rate; (**b**) $r_{2}$ relaxation rate.

**Figure 6 sensors-19-03396-f006:**
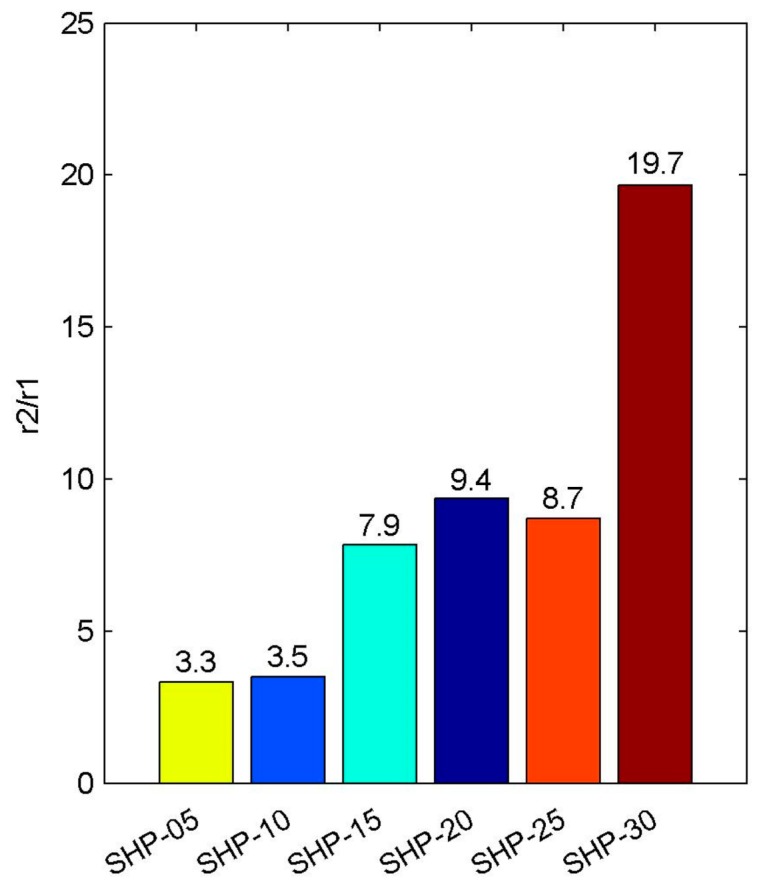
The ratio $r_{2}/r_{1}$ of SHP series magnetic nanoparticles.

**Figure 7 sensors-19-03396-f007:**
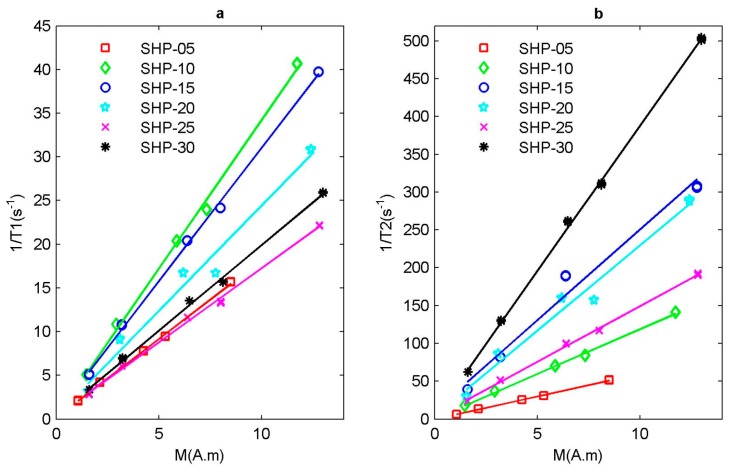
Relaxation rate of SHP series magnetic nanoparticle sample. (**a**) Inverse of longitudinal relaxation time $1/T_{1}$ and (**b**) inverse of transverse relaxation time $1/T_{2}$ with respect to the induced magnetization M.

**Figure 8 sensors-19-03396-f008:**
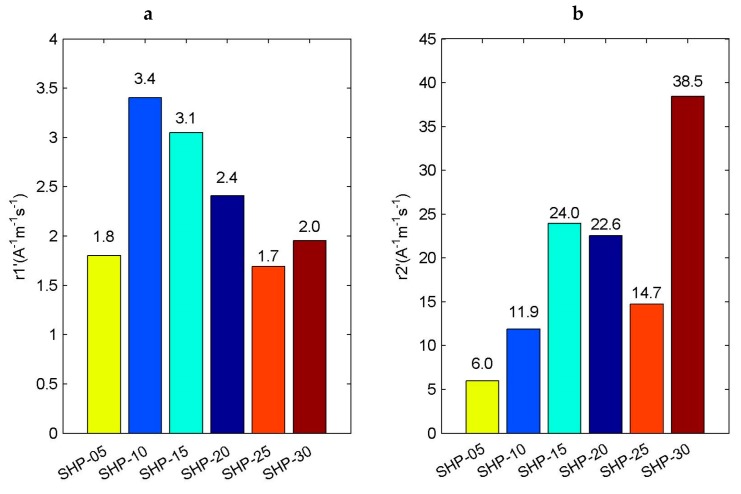
Relaxation rate of SHP series magnetic nanoparticle samples. (**a**) $r_{1}^{\prime }$ relaxation rate; (**b**) $r_{2}^{\prime }$ relaxation rate.

**Figure 9 sensors-19-03396-f009:**
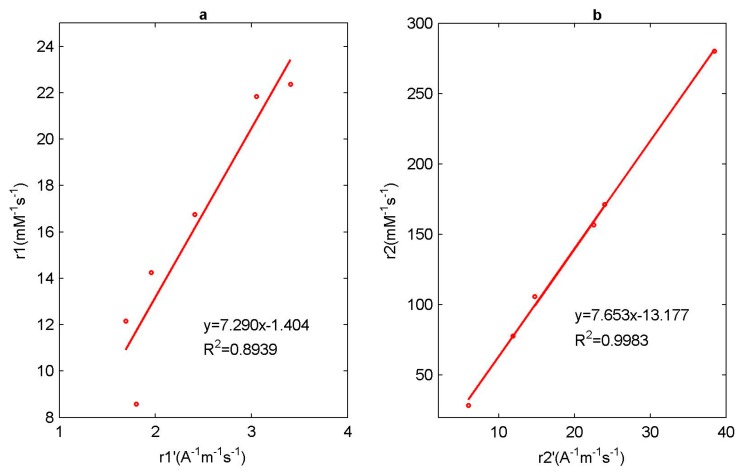
Linear regression of (**a**) $r_{1}$ relaxation rate with $r_{1}^{\prime }$ relaxation rate and (**b**) $r_{2}$ relaxation rate with $r_{2}^{\prime }$ relaxation rate.

**Table 1 sensors-19-03396-t001:** Parameters of lognormal distribution of the hydrodynamic size and the nominal size for SHP series magnetic nanoparticles.

Sample	DLS		Nominal Size (nm)
	Median Diameter (*μ*/nm)	Variance $\left(\sigma^{2}\right)$	
SHP-05	18.24	0.28	5
SHP-10	29.72	0.22	10
SHP-15	37.11	0.24	15
SHP-20	28.22	0.23	20
SHP-25	36.85	0.23	25
SHP-30	40.40	0.24	30

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
