# Peer review of "Characterization and Relaxation Properties of a Series of Monodispersed Magnetic Nanoparticles"

_sensors, 2019, doi:10.3390/s19153396_

Round 1
Reviewer 1 Report
The paper describes the measurement of the relaxation properties of commercial magnetic nanoparticles of different sizes in a half-Tesla nuclear magnetic resonance (NMR) system. The iron oxide particles from Ocean Nanotech with nominal sizes ranging from 5 nm to 30 nm were also characterized by dynamic light scattering and by transmission electron microscopy.
I think the paper is generally well written. The title of the paper is appropriate. In the abstract, the aim of the work is outlined and the results are well summarized. In the introduction, the work is put into proper context to prior publications. In the materials and methods section, the particles and the three measurement modalities TEM, DLS and NMR are briefly mentioned. The transverse relaxation times were examined as a function of the waiting time in the sample chamber and no time dependence was found. Both longitudinal and transverse relaxation times were measured as a function of particle concentration and the usual concentration dependence was obtained. Whereas the individual relaxivities showed no obvious size trend, the ratio r2/r1 of the relaxivities was found to increase with size. The relaxivities were also scaled to the calculated magnetization of the particles. A good correlation was found for transverse relaxivity scaled to concentration and magnetization. In the conclusion, the approach and the main findings are well summarized. Even though there have been many relaxivity studies published thus far, the scaling to magnetization is, to the best of my knowledge, novel and original.
In revision, I suggest to improve the following issues:
Line 18: I found “?1 relaxation rate calculated” and “?2 relaxation rate calculated” a bit confusing. Better write “longitudinal relaxation rate r1” and “transverse relaxation rate r2.” When reading the abstract, I thought that a theory for calculating relaxation rates is being developed, but later I saw that you simply determine the relaxation rates from the relaxation times and concentrations in the standard way. Therefore, I suggest to avoid the word “calculated” here.
Line 139 and Table 1: the parameter µ is the median diameter of a lognormal distribution. The arithmetic mean of Eq. (1) is µ*exp(sigma²/µ).
Line 183: which values did you use for T1,water and T2,water? Did you fit them or use literature values? In the latter case, please give a reference.
Line 202: “It is generally believed that ..”
Line 258: where does this value for Msat come from? Is it a value given by Ocean Nanotech? Or did you take it from the literature (then please cite the reference)?
Lines 260-264: please give details on your “conversion process.” I think this is a really important intermediate step that should be explained to the readers. How did you exactly calculate the magnetization values? Please give the concentration ratios taken from the Ocean Nanotech website in a table, because the values on the website might not be accessible any more in a few years. Please explain the procedure and give your results for the induced magnetizations M of all particles, e.g. in a table.
Line 486: “Available online:”
Reviewer 2 Report
In this manuscript, the authors presented the r1, r2 relaxation rate of monodispersed magnetic nanoparticles using a 0.47T low-field NMR instrument to show the possibility for the application of contrast agents. This manuscript is suitable for publication in Sensors. However, the article needs a major revision for improving the presentation of the results and explanations.
1. The authors only present the TEM result of SHP-05 in figure 1. The TEM images of large-size particles should be presented to proof that the samples are monodispersed magnetic nanoparticles. Besides, the results of DLS shows the hydrodynamic size of SHP series magnetic nanoparticles. SHP-10 and SHP-15 have larger average sizes than SHP-20.However the core size of SHP-20 should be larger than SHP-10 and SHP-15. The authors should explain the results and discuss for that.
2. The authors present the results of concentration-dependence 1/T1 and 1/T2 in figure 4 to obtain the r1 and r2 of relaxation rate of SHP series magnetic nanoparticle samples. Authors should measure more than three independent experiments and present the data as mean value and standard deviation. In addition, it should measure more than 4 concentration values to show that the behavior of the results is linear.
3. In figure 5, the r1 and r2 are independent of the core size of the nanoparticles. The authors should discuss the results.
4. In figure 6. the authors present the r2/r1. The authors should further discuss the trend of r2 / r1 with different core sizes and explain the results.
Round 2
Reviewer 2 Report
The authors have responded appropriately to the comments. The quality of the article has also been considerably improved. It is therefore recommended that this article be published.